# The Prevalence of Dizziness and Vertigo in COVID-19 Patients: A Systematic Review

**DOI:** 10.3390/brainsci12070948

**Published:** 2022-07-20

**Authors:** George Korres, Dimitrios K. Kitsos, Diego Kaski, Anthi Tsogka, Sotirios Giannopoulos, Vasileios Giannopapas, Giorgos Sideris, Giorgos Tyrellis, Konstantine Voumvourakis

**Affiliations:** 12nd ENT Department, Attikon University Hospital, 124 62 Athens, Greece; siderisgior@gmail.com (G.S.); giorgostyrellis91@hotmail.com (G.T.); 22nd Department of Neurology, Attikon University Hospital, 124 62 Athens, Greece; dkitsos@icloud.com (D.K.K.); tsogka.anthi@gmail.com (A.T.); sgianno@med.uoa.gr (S.G.); cvoumvou@outlook.com (K.V.); 3Centre for Vestibular and Behavioural Neuroscience, Department of Clinical and Movement Neurosciences, University College London, 33 Queen Square, London WC1N 3BG, UK; d.kaski@nhs.net; 42nd Department of Neurology, Laboratory of Neuromuscular & Cardiovascular Study of Motion (LANECASM), Attikon University Hospital, 124 62 Athens, Greece; bgiannopapas@gmail.com

**Keywords:** COVID-19, vertigo, dizziness, review

## Abstract

Clinical manifestations of COVID-19 include symptoms of vertigo and dizziness, which is rather unsurprising, since SARS-CoV-2 neurotropism may inflict a broad spectrum of neuropathic effects. The widespread nature of central and peripheral audiovestibular pathways suggests that there may be several probable pathophysiological mechanisms. The cytokine storm, CNS infiltration of the virus through ACE 2 receptors, and other systemic factors can be responsible for the significant number of COVID-19 patients reported to experience symptoms of vertigo and dizziness. In this paper, we present a systematic review of clinical studies reporting the detection of dizziness and vertigo as clinical manifestations of COVID-19 and discuss their etiopathogenesis.

## 1. Introduction

Coronavirus disease 2019 (COVID-2019) was declared a pandemic on the 11 March 2020 by the World Health Organization (WHO) [1], following the previous severe Acute Respiratory Syndrome Coronavirus (SARS-CoV) and Middle East Respiratory Syndrome Coronavirus (MERS-CoV) outbreaks. The epicenter of this pandemic shifted in quick succession from China to Europe to the United States of America and back to Europe in a matter of weeks [2].

Although severe acute respiratory syndrome coronavirus 2 (SARS-CoV-2) mainly affects the respiratory system, neurological complications have been extensively reported [3]. As with SARS-CoV, the COVID-19 virus appears to gain access to the Central Nervous System (CNS) through the angiotensin-converting enzyme 2 (ACE 2) receptor that is expressed in many tissues including the neurological parenchyma.

In this manuscript, we review the reported prevalence of dizziness and vertigo symptoms as part of COVID-19 disease and discuss the potential neurotropic mechanisms of SARS-CoV-2.

### 1.1. The Neurotropism of COVID-19 Disease

The spectrum of neurologic syndromes known to be caused by COVID-19 encompasses encephalitis, meningitis, demyelination and Guillain–Barre Syndrome [4]. amongst others [5]. Direct viral invasion of the brain leading to clinical encephalitis was confirmed by the presence of SARS-CoV-2 in the cerebrospinal fluid (CSF) of patients with COVID-19 through genome sequencing [6].

The inner ear is considered a privileged immunological site due to the absence of lymphatic drainage as well as to the presence of an effective hemato-labyrinthine barrier. The hemato-labyrinthine barrier in the stria vascularis is a highly specialized network that controls ion exchanges between the blood and the interstitial space in the cochlea (and, presumably, also the vestibule), necessary to maintain the ionic gradient and the endocochlear potential for the active processes of mechanical-electrical transduction of the inner and outer hair cells [7]. Local immunity is regulated by the endolymphatic sac, the main antigen-processing site, and its destruction causes a reduction of the immune response in the inner ear [7].

CNS involvement in COVID-19 infection is thought to occur via a hematogenous pathway and retrograde axonal transport [8]. The presence of COVID-19 virus in the general circulation thus facilitates CNS infiltration through ACE 2 receptors expressed in the capillary endothelium. Alternatively, the respiratory entry of the COVID-19 virus to the brain through the cribriform plate close to the olfactory bulb could be another route into the CNS [8], with further virus spread into the brainstem and thalamus [8].

Moreover, the glymphatic system and cerebral microcirculation are likely to allow an interaction between the COVID-19 virus spike protein and the ACE 2 receptors on the capillary endothelium [8]. Subsequent budding of the viral particles from the capillary endothelium and damage to the endothelial lining may favor viral access to the brain [8]. Long before the proposed anticipated neuronal damage occurs, the ruptured endothelium in cerebral capillaries accompanied by bleeding within the cerebral tissue may have fatal consequences in patients with COVID-19 infection [8].

It is well described that severe COVID-19 disease leads to an activation of innate immune cells, also known as cytokine storm, driving a cytokine cell response. Many proinflammatory effector cytokines, such as Tumor Necrosis Factor (TNF), IL 1-β, IL-6, IL-8, have been found in high levels in patients with COVID-19 disease [9], suggesting affinity of the viral spike particle for the ACE 2 receptor [10]. This cytokine production damages the blood–brain barrier (BBB) and may lead to a range of neurological deficits [11].

### 1.2. Vertigo and Dizziness as Part of COVID-19 Manifestations

Vertigo and dizziness have been described among other clinical manifestations of COVID-19, perhaps unsurprisingly, given the anatomically widespread nature of central and peripheral audiovestibular pathways [12]. Thus, SARS-CoV-2 neurotropism may inflict a wide spectrum of neuropathic effects, including effects on neuronal networks governing hearing and balance [12]. Several studies have reported dizziness and vertigo as core clinical manifestations of COVID-19 [12]. Dizziness is the sensation of disturbed or impaired spatial orientation without a false or distorted sense of motion [13]. Vertigo, on the other hand, is a specific type of dizziness defined as the sensation of self-motion when no self-motion is occurring or the sensation of distorted self-motion during an otherwise normal head movement [13]. Dizziness is diagnostically complex and may be attributed to a range of underlying medical conditions such as low blood pressure, cardiac arrhythmias, anemia, and hypoglycemia that may co-exist in patients with COVID-19 disease [14] or may accompany other neurological symptoms as part of a direct neurotropic consequence of COVID-19 [15]. Dizziness and vertigo may also arise as an indirect consequence of COVID-19, such as in the generation of vestibular migraine [16]. In such cases, it is likely that patients have an underlying susceptibility to migraine, with the virus acting as a triggering event, although a thorough medical history of previous migraine or childhood (migrainous) vertigo episodes are often lacking in reported cases. 

Meniere’s disease [17], vascular vertigo [18], and Benign Paroxysmal Positional Vertigo (BPPV) [19] have also been linked with elevated cytokines IL-6 and TNF-α levels, suggesting an inflammatory pathogenesis. In migraine, cytokines and mediators of pain are found in high levels [20]. Studies have shown that arteriosclerosis and resulting reperfusion injury are associated with oxidative stress and inflammatory response, potentially leading to vascular vertigo [18]. These studies indicate that a proinflammatory process and associated immune response to a viral insult may explain the development of vestibular dysfunction following SARS-CoV-2 in patients with genetic susceptibility.

Additionally, acute vertigo in the context of COVID-19 may be attributable to posterior circulation stroke, a recognized risk of this disease, where there is evidence of hypercoagulability [21]. Finally, dizziness and balance disorders can arise from ototoxicity related to the use of antibiotics, such as azithromycin, that have been used in the weaponry against SARS-CoV-2 [22]. Figure 1 outlines the possible pathophysiologic mechanisms of dizziness and vertigo as a result of COVID-19.

Following is a review of published data from original articles, case reports, and open-source datasets to delineate the prevalence of dizziness and vertigo in SARS-CoV-2 cases and the associated diagnostic work-up and treatment used. 

## 2. Methods

We conducted a systematic review of clinical studies reporting the detection of dizziness and vertigo as a clinical manifestation of COVID-19, following PRISMA guidelines [23]. The authors independently performed the literature search, study selection, and data extraction. The PUBMED MEDLINE database was accessed covering the period from 1 January 2020 to 1 January 2022, using the following search terms: dizziness, vertigo, COVID-19, SARS-CoV-2, Coronavirus disease. The retrieved studies from the initial search were further screened for additional articles (Figure 2). 

The following data were extracted from the eligible articles: study characteristics (study title, authors, date of publication, publication type, study site, number of subjects), population characteristics, and the association with neurological disease. Data were extracted independently by four authors (D.K., A.T., V.G. and G.K.). The remaining five authors resolved discrepancies in data extraction and checked the retrieved information to rule out duplications. Regarding quality assessment, two authors (D.K. and A.T.) independently assessed the criteria for the diagnosis of COVID-19, the laboratory confirmation method of SARS-CoV-2, and the respiratory specimens used for testing. Exclusion criteria involved studies that included patients with subjective hearing loss in at least one ear, a previous history of acoustic trauma or prolonged noise exposure, the presence of known audiological pathologies before the diagnosis of COVID-19, previous ear surgery, psychiatric, cardiovascular, or circulatory serious comorbidities, and concurrent or previous medical treatment with chloroquine, aminoglycosides, or azithromycin.

## 3. Results

All included studies regarding dizziness and vertigo in COVID-19 patients are shown in Table 1.

In total, 19 articles were included, of which 14 were observational studies (OS) [24,25,26,27,29,30,33,34,36,37,39,40,41,42], 4 were case reports (CRs) [28,32,35,38], and 1 was a letter to the editor (LtE) [31]. All studies investigated the incidence of dizziness or vertigo as a symptom of SARS-CoV2 infection, and in some studies, data on the diagnostic and therapeutic approaches were included. None of the articles included attempted to provide a definition, as per guidelines referenced above, of either dizziness or vertigo.

All COVID-19 cases reported dizziness or vertigo as a symptom during the infection, which was diagnosed based on retrospective admission data [28,32,35,38]. In all four CRs, dizziness was the initial COVID-19 symptom, and respiratory tract symptoms were manifest at a later stage of the disease [28,32,35,38]. Despite the early presentation of neurological symptoms in the form of dizziness and vertigo in all four CRs, only two of the four cases were thoroughly investigated for these symptoms [32,35], and of those investigated, only one had a brain Magnetic Resonance Imaging (MRI) scan and electronystagmography performed [32]. In the other case, where dizziness or vertigo was investigated, a Computed Tomography (CT) brain scan and bedside clinical examination were the sole diagnostic tests performed [35]. Furthermore, in two of the four cases, where dizziness or vertigo was present, the COVID-19 patients were treated with antihistamines and/or steroids [32,35], and only one of these two cases received vestibular rehabilitation [35]. Maslovara et al. reported two cases of post-BPPV following COVID-19 infection [43]. In 2 of the 14 OSs, dizziness was distinguished from vertigo. Gallus and colleagues reported a dizziness prevalence of 8.3% and a vertigo prevalence of 2% [25]. Viola and colleagues reported a dizziness prevalence of 17.3%, which was the most frequent symptom in a patient subgroup, described as “equilibrium disorders”, and a vertigo prevalence of 1.1% in the same subgroup. The authors also included another subgroup of patients with both tinnitus and equilibrium disorders, without clarifying the exact symptom (7.6%) [40]. Alde and colleagues reported a dizziness prevalence of 16.6%. The authors elaborated further by dividing the “dizziness” subgroup into further patient subgroups with one of the following symptoms: lightheadedness, disequilibrium, pre-syncope, and vertigo (1.66%) [24]. Karadas and colleagues reported a dizziness prevalence of 6.7% and a tinnitus prevalence of 2.1%, without providing any data about vertigo [30]. Karimi-Galougahi presented three COVID-19 PCR-positive cases, of which one case had vertigo and another one had tinnitus [31]. Finally, Lechien and colleagues reported a vertigo prevalence of 0.004% and a tinnitus prevalence of 0.003%, without providing any data on dizziness [33]. For the remaining seven OSs, a dizziness prevalence without further clarifications or definitions was reported. More specifically, Chen and colleagues reported a prevalence of 20% for “dizziness” [26]. Cui and colleagues reported of a prevalence of 20% regarding “Common Otolaryngologic Diagnosis or Symptoms such as dizziness” [27]. Hu and colleagues reported a prevalence of 0.03%, and dizziness was one of the patients’ symptoms described [29]. Lo and colleagues reported a prevalence of 20%, and dizziness was mentioned in the “Symptoms and Signs” section [34]. Mi and colleagues reported a prevalence of 20%, and dizziness was mentioned in the “Symptoms and Signs” section [37]. Vacchiano and colleagues reported a prevalence of 10%, describing dizziness as a “Typical Neurological Symptom” [39]. Mao and colleagues reported a prevalence of 16.8%, mentioning dizziness as a common neurological manifestation [36]. Finally, Wang and colleagues reported a prevalence of 7% and 9.4% for dizziness and vertigo, respectively, while dizziness was mentioned as a “more common symptom in patients admitted to intensive care unit” [41,42].

## 4. Discussion

Neurological symptoms have been frequently observed with COVID-19 infection, many of which are not specific to SARS-CoV-2, being associated with many other viral infections [44]. Dizziness and vertigo seem to be prevalent symptoms in COVID-19 patients and, as non-specific symptoms, may be overlooked in patients with additional neurological symptoms [44].

Dizziness and vertigo may be the direct consequence of viral disease affecting the vestibular function along the virus widespread course (from the inner ear to the brain) [45]. Moreover, dizziness (particularly) and vertigo may occur indirectly due to respiratory distress, hypoxia, hypotonia, dehydration, and fever during sepsis, for example. These mechanisms of neuropathogenicity may be sufficient to explain headache and dizziness as frequent non-specific symptoms in mild or moderate COVID-19 patients [45]. Dizziness may go unrecognized because impaired consciousness is frequently noted in hospitalized elderly patients [46].

In the single case reported by Karimi-Galougahi, dizziness and vertigo was argued to be a direct consequence of COVID-19 infection, accompanied by unilateral hearing loss [31]. Alde and colleagues concluded that most cases of dizziness were attributable to lightheadedness, which was probably exacerbated by psychophysical stress following acute infection and mandatory quarantine [24]. Karadas and colleagues concluded, from their study on a large 239-patient cohort, that dizziness and vertigo are common and early symptoms of COVID-19 that could aid a prompt diagnosis and potentially prevent viral spread [30]. Mao and colleagues argued that dizziness is the most common neurological manifestation of COVID-19 [36], although this may reflect the non-specific nature of the symptom rather than a specific affectation of the inner ear or of central vestibular structures. Viola and colleagues, however, suggest that hearing and balance dysfunction may be directly related to inner ear vascular damage, because the labyrinth is particularly susceptible to ischemia due to the characteristics of terminal vasculature of its vessels and its high energy requirement. They further propose that prolonged hospitalization and bed rest may be responsible for otolith detachment [40], presumably leading to BPPV and otolithic disturbance, although this has not been formally demonstrated. An important conclusion made by Chen and colleagues is that dizziness and vertigo typically accompanied other common non-specific neurological symptoms of COVID-19 infection, including headache [26], arguing that dizziness and vertigo may not be directly related to a direct neurotropic damage to vestibular structures. 

We identified a prevalence between 0.03% and 20% for dizziness and between 0.004% and 12% for vertigo across all observational and cohort studies included. None of the OSs, CRs, or LtE used any dizziness or vertigo definitions as per guidelines. Hence, in many OSs, the spectrum of symptoms that were described as “dizziness” or “vertigo” differed from those of established definitions, perhaps accounting for such a wide range of prevalence being reported. As a result, moreover, there was a funnel asymmetry detected visually and quantitatively in the pooled analyses of “dizziness” and “vertigo” symptoms, which could imply the presence of publication bias. In addition, there was a substantial heterogeneity between studies, perhaps related to variations in age, disease severity, and underlying comorbidities that were difficult to subgroup due to limited information on the available data. In addition, the age range of the population enrolled in the studies was wide, and this led to variance in the incidence of many conditions that could contribute to dizziness and vertigo. Many patients across the included studies had a broad spectrum of co-morbid conditions which might have affected the emergence of neurological symptoms, including dizziness and vertigo, complications, and outcomes. In addition, mild COVID-19 cases were substantially overrepresented in these studies compared to severe/critical cases, where neurological symptoms like dizziness were usually overlooked [46].

A key limiting factor in understanding the relationship between dizziness and vertigo and COVID-19 is the failure of many studies to describe in detail the diagnostic process and management of dizziness and vertigo symptoms.

To date, dizziness and to a lesser extend vertigo, are nevertheless considered core COVID-19 non-respiratory symptoms, and their presence may aid a prompter diagnosis. Both symptoms, however, require thorough investigation to determine their underlying cause, that may include peripheral syndromes such as acute labyrinthitis, vestibular neuritis, acute otitis media, or central disorders such as vestibular migraine and stroke. Similarly, vertigo symptomatology needs to be associated, wherever possible, with other audiovestibular manifestations, such as hearing loss and tinnitus. Finally, persistent dizziness or vertigo after convalescence from COVID-19 requires specialist referral for detailed investigation and management. 

## 5. Conclusions

In this review, we sought to identify the incidence of dizziness and vertigo among SARS-CoV-2-infected patients. Large prospective longitudinal studies should be undertaken to carefully document the incidence of dizziness and vertigo, as well as associated complications and outcomes, over the course of this illness. Additionally, more data are needed to decipher the pathogenesis of COVID-19-associated vertigo/dizziness and understand whether they represent a consequence of direct viral insult, the effects of proinflammatory cytokines on the CNS, or perhaps indirect consequences of systemic illness. To this end, investigations including measurements of cytokine levels and neuro-otological tests such as videonystagmography, objective recording of the vestibulo-ocular reflex, Pure Tone Audiometry (PTA), and cervical Vestibular Evoked Myogenic Potentials (cVEMPs) should be conducted, and neurologists and otorhinolaryngologists should be consulted early to better define their underlying cause and instigate appropriate management.

## Figures and Tables

**Figure 1 brainsci-12-00948-f001:**
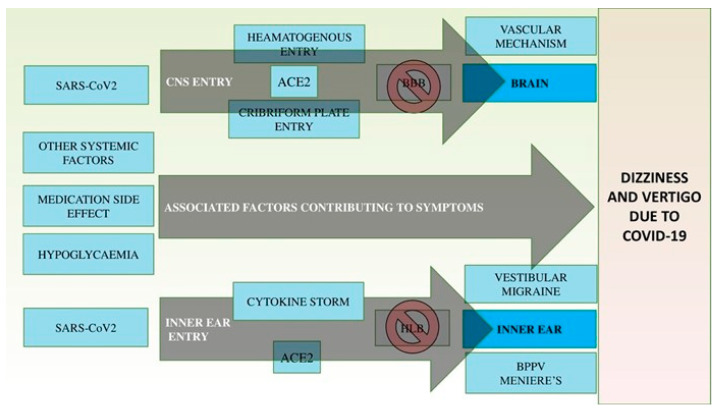
COVID-19 neurotropic and inner ear mechanism of action. SARS-CoV2: severe acute respiratory syndrome coronavirus 2, CNS: Central Nervous System, ACE2: Angiotensin-Converting Enzyme 2, BBB: Blood Brain Barrier, HLB: Hematogenous Labyrinthine Barrier, BPPV: Benign Paroxysmal Positional Vertigo.

**Figure 2 brainsci-12-00948-f002:**
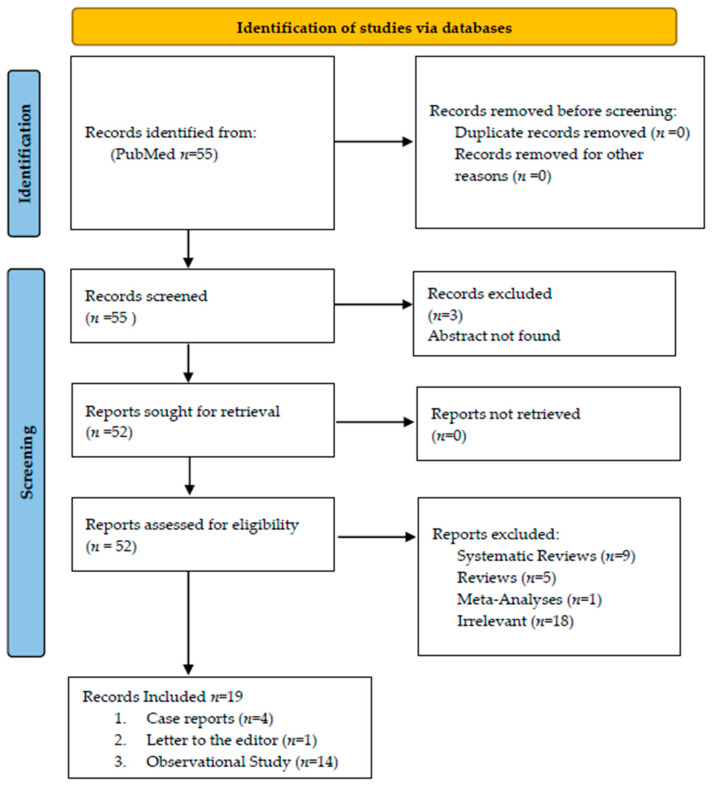
PRISMA flow chart.

**Table 1 brainsci-12-00948-t001:** COVID-19 studies including dizziness and vertigo.

Author	Study Type	Sample Size	Dizziness	Vertigo	Other
Alde et al., 2021 [24]	OS	1512	-	12% (30/251)	Lightheadedness 43.8% (110/251) Disequilibrium 27.9% (70/251) Presyncope 16.3% (41/251)
Gallus et al., 2021 [25]	OS	48	3% (4/48)	2% (1/48) Spinning Vertigo	Tinnitus 4,2% (2/48) Dynamic imbalance 2% (1/48) Static imbalance 6% (3/48)
Chen et al., 2020 [26]	OS	145	20% (29/145)	-	-
Cui et al., 2020 [27]	OS	20	20%(5/20)	-	-
Han et al., 2020 [28]	CR	1	-	√	-
Hu et al., 2020 [29]	OS	24	0.03% (1/28)	-	-
Karadas et al., 2020 [30]	OS	239	6.7% (16/239)	-	Tinnitus 2.1% (5/239) Neurological findings 34% (83/239)
Karimi-Galougahi et al., 2020 [31]	LtE	6	-	33% (1/3)	Tinnitus (3/3) Unilateral hearing loss (3/3)
Kong et al. 2020 [32]	CR	1 Electronystagmography, Pure Tone Audiometry, Brain MRI performed. Symptomatic Treatment given	√	-	-
Lechien et al., 2020 [33]	OS	1420	-	0.004% (6/1420) Rotatory Vertigo	Tinnitus 0.003% (5/1420)
Lo et al., 2020 [34]	OS	10	20% (2/10)	-	-
Malayala et al., 2020 [35]	CR	1 patient Suspected COVID-19-induced acute vestibular neuritis. CT brain performed Symptomatic Treatment given Vestibular Rehabilitation performed	-	√	-
Mao et al., 2020 [36]	OS	214	16.8% (36/214)	-	-
Mi et al., 2020 [37]	OS	10	3% (3/10)	-	-
Sia et al., 2020 [38]	CR	1	Sudden onset	-	-
Vacchiano et al., 2020 [39]	OS	133	10%(11/133)	-	-
Viola et al., 2020 [40]	OS	185	94.1% (32/34)	5.9% (2/34) acute vertigo attacks	Equilibrium disorders 18.4% (34/185) Tinnitus 23.2% (43/185) Combination 7.65% 14/185 (tinnitus and equilibrium disorder)
Wang et al., 2020 [41]	OS	69	5/69 (7%)	-	-
Wang et al., 2020 [42]	OS	138	13/138 (9.4%)	-	-
Maslovara et al., 2021 [43]	CR	2	-	BPPV	-

PCR: Polymerase Chain Reaction, ICU: Intensive Care Unit, CR: case report, OS: observational study, LtE, letter to the editor, BPPV: Benign Paroxysmal Positional Vertigo.

## Data Availability

Not applicable.

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
