# Peer review of "The Prevalence of Dizziness and Vertigo in COVID-19 Patients: A Systematic Review"

_brainsci, 2022, doi:10.3390/brainsci12070948_

Round 1

Reviewer 1 Report

The paper is excellently written, and much effort has been invested in data collection and meta-analysis. I am only sorry that the authors did not refer to one recently published CR on Post-COVID-19 BPPV: Maslovara, Siniša, and Andro Košec. “Post-COVID-19 Benign Paroxysmal Positional Vertigo.” Case reports in medicine vol. 2021 9967555. 1 Jun. 2021, doi: 10.1155 / 2021/9967555

Author Response

Many thanks for your kind words, and for pointing that paper out, the reference is now included in our review

Reviewer 2 Report

Very good work! 

Author Response

Many thanks for your kind words

Reviewer 3 Report

1) I would suggest improving the summary table and the result section.

2) Which was the main goal of the review? Investigate the prevalence?

3) I would divide the table as follow:

I would collect the observational studies with their prevalence of dizziness, unbalance, and true vertigo.

Thus, I would collect case reports and case series: for these study types would be interesting to specify the subtype of vertigo, the investigations performed (pure tone audiometry, VNG, vHIT, imaging) and the treatment.

4) What does mean no data on COVID 19 patients with vertigo? If no specific data is reported, these studies are probably not eligible for the review. Please specify.

Moreover, it would be interesting to understand how many specific diagnoses of vertigo (neurological, audiovestibular…) are reported.

Author Response

Comment 1

I would suggest improving the summary table and the result section.”

Reply

Thank you for your valuable comment.
We have changed the table format and split the dizziness, vertigo and other symptoms prevalence.

Comment 2

Which was the main goal of the review? Investigate the prevalence?”

Reply
The main goal of our review was to report literature cases of dizziness and vertigo regarding their prevalence of such symptoms in Covid-19 patients, and to provide a hypothesis regarding the possible pathophysiology of vestibular symptoms in patients with Covid-19. This has been clarified in the revised manuscript thus:

“In this manuscript, we review the reported prevalence of dizziness and vertigo symptoms as part of COVID 19 disease and discuss the potential neurotropic mecha-nisms of SARS-CoV-2.”

Comment 3

I would divide the table as follow: I would collect the observational studies with their prevalence of dizziness, unbalance, and true vertigo. Thus, I would collect case reports and case series: for these study types would be interesting to specify the subtype of vertigo, the investigations performed (pure tone audiometry, VNG, vHIT, imaging) and the treatment.”

Reply

Thank you for your comment.
We have now changed the table format and divided the prevalence of dizziness, vertigo and other symptoms. Unfortunately, there are only limited papers that include tests and investigations results relevant to symptoms of dizziness and vertigo, perhaps because at the time these were not considered to be life-threatening symptoms.

Comment 4

“What does mean no data on COVID 19 patients with vertigo? If no specific data is reported, these studies are probably not eligible for the review. Please specify. Moreover, it would be interesting to understand how many specific diagnoses of vertigo (neurological, audiovestibular…) are reported”

Reply

Thank you for your helpful comment. We agree that our statement ‘No data on COVID-19 cases with vertigo’ is ambiguous and has therefore now been removed from the table. Most of the studies available, that were also included in our systematic review did not provide an accurate definition of dizziness and/or vertigo and the vast majority of them did not corroborate the diagnostic tools and batteries that were used to make a specific diagnosis. This is a major limitation of the available evidence and has been highlighted in the review:“None of the OSs, CRs or LtE used any dizziness or vertigo definitions as per guidelines.”

And

“A key limiting factor in understanding the relationship between dizziness and vertigo and COVID-19 is the failure of many studies to describe in detail the diagnostic process and management of dizziness and vertigo symptoms.”